# Point Cloud Repair Method via Convex Set Theory

**Tianzhen Dong, Yi Zhang \*, Mengying Li and Yuntao Bai**

Visual Intelligence Perception Laboratory, School of Computer Science and Information Engineering, Shanghai Institute of Technology, Shanghai 201418, China
* Correspondence: 206141138@mail.sit.edu.cn

**Abstract:** The point cloud is the basis for 3D object surface reconstruction. An incomplete point cloud significantly reduces the accuracy of downstream work such as 3D object reconstruction and recognition. Therefore, point-cloud repair is indispensable work. However, the original shape of the point cloud is difficult to restore due to the uncertainty of the position of the new filling point. Considering the advantages of the convex set in dealing with uncertainty problems, we propose a point-cloud repair method via a convex set that transforms a point-cloud repair problem into a construction problem of the convex set. The core idea of the proposed method is to discretize the hole boundary area into multiple subunits and add new 3D points to the specific subunit according to the construction properties of the convex set. Specific subunits must be located in the hole area. For the selection of the specific subunit, we introduced Markov random fields (MRF) to transform them into the maximal a posteriori (MAP) estimation problem of random field labels. Variational inference was used to approximate MAP and calculate the specific subunit that needed to add new points. Our method iteratively selects specific subunits and adds new filling points. With the increasing number of iterations, the specific subunits gradually move to the center of the hole region until the hole is completely repaired. The quantitative and qualitative results of the experiments demonstrate that our method was superior to the compared method.

**Keywords:** point cloud; convex set; Markov random field; variational inference

## 1. Introduction

The point cloud is an important means to describe the surface shape of 3D objects. Limited by environmental factors or scanning equipment, a collected 3D point cloud is incomplete, and incomplete point-cloud data reduce the accuracy of downstream work, such as 3D target reconstruction, target recognition, and scene tracking and understanding. Therefore, point-cloud repair is indispensable work in 3D vision. Figure 1 shows a point cloud with holes.

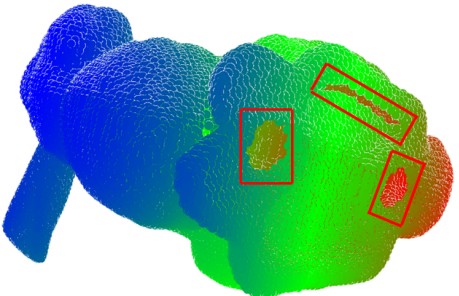

**Figure 1.** Point cloud with holes.

The disorder of space points renders point-cloud repair more challenging. Some current methods [1–3] established a mesh model and repaired the shape of the object surface via the mesh model. The purpose of meshing is to mitigate the influence of the

disorder of the point-cloud data. Harary et al. [4] utilized the mesh model for repairing. The characteristic curve was restored, and the divided smooth subhole area was repaired. Nonetheless, this method needs to manually select the points needed for fitting the curve, and it cannot adaptively select the required points. The construction of the mesh is a complex process, and the quality of mesh distribution directly reduces the repair effect.

There are some methods to repair point clouds through block matching. Fu et al. [5] transformed the repair problem into an optimization problem using nonlocal self-similarity and a local smoothing constraint to achieve better point-cloud repair. Sun et al. [6] proposed a data-driven point-cloud completion method that repairs the point cloud through symmetric relationship. The dataset was used to infer the 3D shape in the absence of a symmetrical relationship. This method could repair some point-cloud models with serious missing areas, but the effect of repairing irregular objects is poor. Gregor et al. [7] needed to ensure that there were many symmetrical or repeated structures in the visible region; however, the conditions required by this method were relatively strict, so the scope of application was limited. The effectiveness of block-matching methods usually depends on the large number of models and rich model types in the database.

With the successful application of PointNet [8] and PointNet++ [9] on point clouds, the method based on depth learning [10–17] was applied to point-cloud completion. Point-cloud completion methods implemented with an encoder–decoder framework mainly follow the coarse-to-fine principle to achieve point-cloud completion [18,19]. However, such methods often add new points in the nonhole area, changing the inherent structure information of the point cloud [20–23]. At the same time, the effectiveness of these methods is usually limited by the size of the training data [24,25].

To solve these problems, we propose a novel point-cloud repair method that transforms the point-cloud repair problem into a problem of constructing a convex set. In this paper, we weaken the convex set's general form, so that it could be used in point-cloud repair.

In this paper, we weakened the convex set's general form, introduced the MRF to predict the special subunit, and added new 3D points in the specific subunit, so that the subunit would become a convex set. First, we built a 3D unit S with the size of $L \times W \times H$ in the neighborhood of the hole boundary (S had to contain $T$ adjacent boundary points, and the value of T was positively related to the density of the boundary neighborhood. In this paper, $T \geqslant 3$). Second, we limited the distribution of new filling points according to weakened convex set theory, so that new spatial points were distributed in the hole area. The calculation of the distribution area of new filling points is mainly divided into two stages, namely, coarse screening and fine screening. At the coarse-screening stage, we discretized S into K subunits and set the label of the subunits with 3D points to 1; otherwise, we set it to 0, thus obtaining the initial label field. Then, the subunits without filling points were eliminated by using the neighborhood attribute information of boundary points to reduce the cost of subsequent calculation. Fine screening is used to accurately predict subunits that need new filling points. Therefore, we introduced a Markov random field (MRF) to transform the selection of subunits into the maximal a posteriori probability (MAP) of the random field label and then used the variational mean field to approximate MAP. Lastly, we added new points to the final filtered subunit, so that the specific subunit would become a convex set.

We reconstructed the original features of the point-cloud surface by constructing a convex set of subunits located in the missing area. The division of the subunit set and the construction of the convex set were carried out iteratively. Each iteration made the subunit gradually move towards the direction of hole shrinkage and lastly repair the entire hole area. Our method could accurately infer the distribution of new filling points to ensure that the repaired shape was as consistent with the real surface shape as possible. The proposed method is not limited by the category of point clouds and could repair various categories of point clouds while maintaining the position of points in nonhole areas unchanged. The key contributions of this paper are as follows:

(1) We introduce a Markov random field and transform the subunit selection problem into the maximal a posteriori of the random field label. By solving the maximal probability, we ensure that the new filling point is always located in the missing area.

(2) We introduce and weaken the concept of a convex set, and redefine the problem of point-cloud repair into the problem of the composition of a convex set. The entire hole area is gradually repaired by constructing the specific subunit into a convex set.

## 2. Methods

The input of our method is a 3D point cloud with holes, and the output is a fully repaired point cloud. The main flow of our algorithm is shown in Figure 2. The extraction of boundary points is the reference centroid method [26,27].

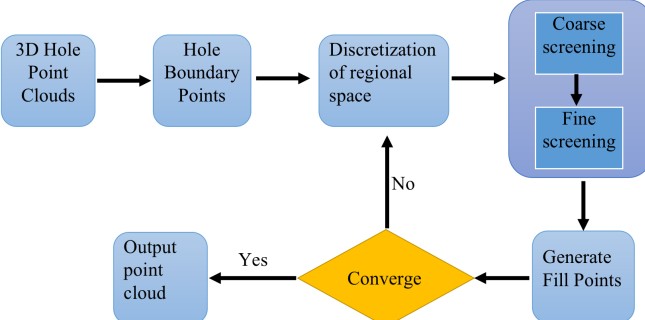

**Figure 2.** The overview of our method.

### 2.1. Hole Definition

Let $\varphi_1$ be the missing area inside the 3D point cloud, and $\varphi_2$ be an unclosed missing area at the outer boundary. As illustrated in Figure 3, the line segment connecting $A$ and $B$ is marked by $L_{AB}$, where $A$ and $B$ are points on the $\varphi_2$ boundary. Any boundary point in $\varphi_2$ is in the closed area defined by line segments $L_{AB}$ and $\varphi_2$. Assuming that $0 \leq \Delta \leq 0.1$ (in this paper, $\Delta = 0.1$), the boundary arc length of $\varphi_2$ is $C_\varphi$. If $L_{AB} < \Delta C_\varphi$, $\varphi_2$ is considered a boundary hole. Inner hole $\varphi_1$ and boundary hole $\varphi_2$ are collectively referred to as *hole*.

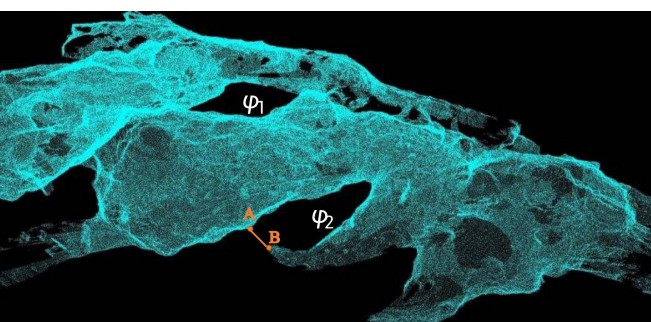

**Figure 3.** Example of point-cloud holes.

### 2.2. Weakening of Convex Set

Because the position of the 3D point is uncertain, a looser uncertainty analytical method is needed to deal with this ill-posed problem. Nonprobabilistic set theory can effectively deal with various uncertainty problems. If a set is convex, it is called a convex set. In the 3D point cloud, to reasonably distribute new filling points, we weakened the general definition of the convex set, so that it could be used in point-cloud repair and avoid the confusion of point distribution due to the uncertainty of the 3D point.

**Definition 1.** *In the repair of a 3D point cloud, let S be a 3D vector space containing $x_i$ and $x_j$. For the segment formed by the connection of $x_i$ and $x_j$ in subset s of S, when all new filling points contained in s are in the direction domain of the line segment, s is a convex set. As shown in Figure 4,*

*the new fill (blue) points were located in a direction of the line segment connected by $x_i$ and $x_j$. In the repair of 3D point clouds, the properties of convex sets are as follows: (1) The intersection of any convex set is also a convex set.*

*(2) If s is a convex set, any new filling point $x_{new}$ in s is located on the side of the line segment formed by $x_i$ and $x_j$.*

Considering the gradualness of the spatial point-cloud data change, it was assumed that the point-cloud data of the area to be filled and the hole edge point cloud had similar distribution rules, and the distribution of the points in the area to be filled could be inferred from the geometric information of the edge point set. When the set consisting of new filling points and edge points in a specific area meet the appeal definition, the set is convex.

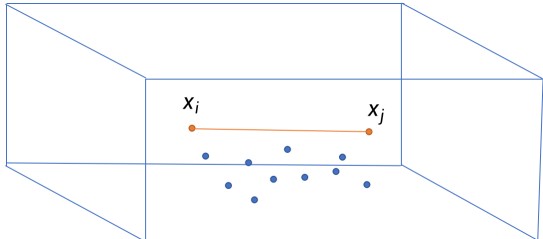

**Figure 4.** Convex set formed by new filling points.

### 2.3. Discretization of Regional Space

In general, the smaller the hole area is, the higher the repair accuracy. In view of this, we discretized the 3D space around the hole boundary points and divided it into k subunits. We defined $X$ as the set of boundary points. $X = \{x_1, x_2, \cdots, x_m\}$, $m$ is the number of hole boundary points. We first selected $T$ ($T \geqslant 3$) adjacent boundary points $x_i, \cdots, x_{i+(T-1)}$ and took midpoint $\hat{x}_i$ of $x_i$ and $x_{i+(T-1)}$ as the base point to define 3D space $S$ with a size of $L \times W \times H$, and $\hat{x}_i$ was the central point of $S$. $x_{i+(T-1)}$ is denoted by $x_j$ for convenience in the description. Then, we discretized 3D space $S$, so that the resolution of each spatial unit was $d_L \times d_W \times d_H$. After discretization, the 3D tensor of shape $\frac{L}{d_L} \times \frac{W}{d_W} \times \frac{H}{d_H}$ was obtained. Because we needed to include $x_i$ and $x_j$ in the spatial unit, we took $d_L = L = ||x_i - x_j||^2$ and divided $S$ into $k$ ($k = 9$) subunits, $S = \{s_1, s_2, \cdots, s_k\}$. After discretization, the 3D points contained in $S$ were also allocated to different subunits. We took subunit-containing points $x_i$ and $x_j$ as the central unit $s_1$, and the subunits surrounding the central unit were neighborhood subunits $s_k(s_k \neq 1)$. With the increasing number of iterations, the specific subunits gradually moved to the center of the hole region until the hole had been completely repaired.

### 2.4. Filtering of Spatial Subunits

In general, new filling points in the hole area and 3D points in the nonhole area should be distributed on both sides of the hole boundary. For this reason, we assumed that there was a special subunit $s$ in $S$ and $s$ was located in the hole area. When $s$ is a convex set, the 3D points in $s$ all meet Property 2 of the weakened convex set, which means that the filling points in the $s$ are distributed in the hole area.

Specifically, the density of 3D points contained in subunit $s_k(s_k \subset S)$ may have three conditions: (1) the density of 3D points is close to $\rho$; (2) the density of points is less than $\rho$; (3) the density of the point is 0. $\rho$ is the mean density of the points in neighborhoods $x_i$ and $x_j$. According to the experimental analysis, the probability that Condition 2 would occur in central subunit $s_1$ was the highest. However, Condition 3 only appeared in subunit $s_k(s_k \neq 1)$, so it was only necessary to filter the unit blocks in Condition 3 to obtain the subunit to which the filling points had to be be added and record it as $s^*$. We defined $s_1$ and $s^*$ to form a specific subunit $s$. The selection of $s$ is mainly divided into two steps: coarse screening and fine screening.

### 2.4.1. Coarse Screening

We used the similarity feature of the normal point-cloud neighborhood vector to conduct coarse screening. First, we used the point set of the neighborhood of point $\hat{x}_i$ ($\hat{x}_i$ is the midpoint of $x_i$ and $x_j$) to calculate normal vector $\vec{n}$. Considering that 3D spatial unit $S$ on different scales contains a different number of points, it is impossible to accurately calculate the normal vector of the surface composed of $x_i$ and the set of neighborhood points. To solve this problem, we calculated normal vector $\vec{n}_i$ at three scales (the number of points on the three scales was 7, 14, and 21) and mean value $\hat{\vec{n}}_i$ as the contrast value. Then, the surface normal vector formed by the central point of each subunit and the nearest neighbor point of $k$ ($k$ was 7, 14, or 21), and mean value $\sigma$ were calculated. Lastly, we used cosine similarity to calculate the difference between normal vector $\vec{\sigma}$ and contrast value $\hat{\vec{n}}_i$. The cosine similarity was calculated as follows:

$$cos\left(\vec{\sigma}, \widehat{\vec{n}}\right) = \frac{\sum_{i=1}^{3}\left(\vec{\sigma}_i \times \widehat{\vec{n}}_i\right)}{\sqrt{\sum_{i=1}^{3}\left(\vec{\sigma}_i\right)^2} \times \sqrt{\sum_{i=1}^{3}\left(\widehat{\vec{n}}_i\right)^2}}. \tag{1}$$

We needed to specify a rule to retain those spatial cells with high normal vector similarity. Therefore, we calculated the degree of difference of any two vectors in $\vec{n}_i$, and found the maximal $cos_{max}$ and took it as the threshold. if $cos(\vec{\sigma}, \vec{n}_i) \geqslant \vartheta * cos_{max}$, then we reserved the subunit. $\vartheta$ is a controllable parameter. In this paper, $0.50 \leqslant \vartheta \leqslant 1.00$. Coarse screening eliminates some useless subunits, reducing the calculation amount for the subsequent fine screening.

### 2.4.2. Fine Screening

To ensure that the repaired point-cloud surface was closer to the real surface, the filtering results needed to be further refined. The Markov random field (MRF) [28] is an undirected probability graph model that is a random field with Markovian characteristics. The value of each node is only related to the surrounding nodes. The K-nearest neighbor (K-NN) graph of an unordered point cloud is an undirected graph constructed by connecting each point with its nearest K neighbors [5]. Therefore, we introduced MRF, took the geometric attributes of local point clouds as the observational information, and transformed the fine screening of elements into the maximal a posteriori (MAP) of random field labeling. Given an incomplete 3D point-cloud model $P$, let $G(V, E)$ represent MRF, where $V$ represents a set of voxels, and $E$ represents a set of edges connecting subunits. Each subunit $v_i \in V$ was assigned a label $l_i \in \{0, 1\}$. The units retained after rough screening were regarded as an unstable label set, and recorded as Label 0. Similarly, the units with 3D points in their neighborhood were recorded as 1. Let $L = \{l_i\} \in \{0, 1\}^{|V|}$ be a set of labels. We optimized the energy of the subunits with $l_i = 0$; that is, we determined whether the subunit needed filling points. First, the state distribution function of the MRF was obtained according to the Hammersley–Clifford theorem:

$$P(L) = Z^{-1} \prod_{Q \in C} \psi_Q(L_Q), \tag{2}$$

where $z$ is the normalization factor to ensure that $P(L)$ forms probability distribution. $Q$ is a subgroup, $C$ is a group set, and $\psi_Q$ is the potential energy function corresponding to group $Q$ that is used to model the variable relationship in group $Q$. To ensure the non-negativity of the potential energy function, $\psi_Q$ is defined as follows:

$$\psi_Q(L_Q) = exp^{-\left\{\alpha_i \sum_{i \in V} V_i(l_i) + \beta_{i,j} \sum_{i,j \in E} V_{i,j}(l_i, l_j)\right\}}, \tag{3}$$

where $\alpha_i$ and $\beta_{(i,j)}$ are adjustable parameters. The former only considers the potential energy of a single node, while the latter considers the relationship between each pair of nodes. $V_{i,j}(l_i, l_j)$ represents the cost of placing labels $l_i, l_j$ at two adjacent subunits $i, j$:

$$V_{i,j}(l_i, l_j) = exp^{1 - \frac{1}{1 + cos(n_i, n_j) + \alpha}}, \tag{4}$$

where $\alpha = 0.000001$ to avoid 0 denominators, and $n_i$ and $n_j$ are the normal surface vectors formed by the subunit's central point and the set of neighboring points. Similarly, $V_i(l_i)$ represents the observational cost of a single cuboid unit $i$ in each state $l_i$.

$$V_i(l_i) = logP(n_i|l_i). \tag{5}$$

Like the traditional Markov random field, likelihood function $P(n|l)$ is established as follows:

$$P(n|l) = \prod_{i=1}^{N^1} P(n_i|l_i) = \prod_{i=1}^{N^1} \frac{1}{\sqrt{2\pi}\sigma_{l_i}} \prod_{i=1}^{N^1} \frac{1}{\sqrt{2\pi}\mu_{l_i}}. \tag{6}$$

where $P(n|l)$ represents the joint probability of observational variable $n$ when the state $l$ of MRF is given. When building the label field model of MRF, we used $M$ Gaussian distributions to fit the histogram distribution of the normal vector information of the surface formed by the central point of the unstable subunit and the neighboring point set. $M$ indicates that labels are divided into several categories. In this paper, labels were divided into two categories, $M = 2$. The expectation maximization (EM) algorithm was used to calculate the mean $\mu = \{\mu_1, \mu_2, \cdots \mu_M\}$ of the M Gaussian distributions and standard deviation $\sigma = \{\sigma_1, \sigma_2, \cdots \sigma_M\}$. The detailed solution process of the likelihood function can be found in [29]. According to the obtained likelihood function $P(n|l)$ and the state distribution $P(L)$ of the MRF, the optimal state of the MRF of the subunit set is as follows:

$$L^* = maxP(l|n) \propto max\{P(n|l)P(L)\}. \tag{7}$$

According to Equation (7), we transformed the problem of solving the optimal state of the MRF of the subunit set into a problem of calculating the maximal a posteriori (MAP) estimate. Variational inference (VI) is a large class of Bayesian approximate inference methods that can transform a posteriori inference problems into optimization problems for a solution. The core idea of variational inference is to maximize objective function $J(Q)$ to produce variable distribution $Q(l)$. $Q(l)$ is the mean field in the form of $\prod_{i=1}^{|M|} Q_i(l_i)$. The objective function $J(Q)$ is defined as follows:

$$J(Q) = \sum_l Q(l)logP(l, n) - \sum_l Q(l)logQ(l). \tag{8}$$

We could calculate by substituting $Q(l)$ into Equation (8):

$$J(Q) = -KL(Q(l)||P(l|n)) + const. \tag{9}$$

KL divergence is used to measure the similarity between $Q(l)$ and target $P(l|n)$, KL is non-negative, and const is a constant. $J(Q)$ is essentially the evidence lower bound objective (ELBO), which is a function of $Q$. According to Equation (9), calculating $argmax\{J(Q)\}$ is equivalent to minimizing KL. In other words, we needed to find a posteriori $Q^*(l)$, so that $Q^*(l) = L^*$. In this way, the problem of variational inference is transformed into an optimization problem. The solution of maximal $J(Q)$ satisfies:

$$logQ^*(l_j) = E_{\prod_{i \neq j} Q_i(l_i)}\{logP(l, n)\}, \tag{10}$$

where $E_{\prod_{i \neq j} Q_i(l_i)}\{\cdot\}$ is the expectation about $Q(l)$. When solving, $Q^*(l_j)$ is fixed first and then $Q^*(l_j)$ is updated with $logP(l, n)$ on $\prod_{i \neq j} Q_i(l_i)$. There were several iterations until

$logQ^*(l_j)$ had converged to a fixed value, and the maximal $J(Q)$ had been obtained. The optimal state after the solution was to allocate unstable subunits to stable label set ($l_i = 1$). When the label of a pointless subunit is 1, the spatial unit is determined to be filled with new points.

### 2.5. Generate Fill Points

After twice screening, subunit $s^*$ with the highest probability in space $S$ around $x_i$ and $x_j$ was calculated; $s^*$ and central subunit $s_1$ together formed a specific subunit $s$ ($s$ includes the boundary area and the hole area). Then, we needed to add new points to the subunit $s$. Let the direction from $x_i$ to $x_j$ be $n_{i,j}$ according to the definition of the weakened convex set when all the new filling points are in the closed area contained by $s$, all the new points are in the direction domain of line segment $C_{i,j}$ connected by $x_i$ and $x_j$, $s$ is a convex set. The filling direction of the new point is:

$$v = \frac{\widehat{\overline{n}} \times n_{i,j}}{||\widehat{\overline{n}} \times n_{i,j}||_2^2},$$

(11)

where $v$ is a normalized vector pointing in the direction of hole shrinkage. By combining the Markov random field and convex set theory, we determined the distribution area of new points (along the direction of hole shrinkage). In combination with density $\rho$ of the neighborhood point set, enough points were uniformly sampled on segment $C_{i,j}$ and new points were added along the direction of $v$:

$$x_{new} = q + \zeta v,$$

(12)

where $q$ is a point on $C_{i,j}$, and $\zeta$ is a controllable parameter to ensure that $x_{new}$ is in the subunit. After generating enough 3D points in specific subunit $s$, $x_i$ was moved out of the $X$ set and continued to iterate $x_{i+1}$ and $x_{j+1}$ in the same way. When there is only one element left in $X$, and the hole boundary is closed, the last element should pair with the first element in $X$ to form a unit $S$. When $X$ was an empty set, we added the boundary formed by the new 3D points to $X$ to generate new fill points again. During the whole repair process, the specific subunit gradually moved to the center of the hole region until the hole had been completely repaired.

## 3. Results and Discussion

In this section, we experimentally analyze the method proposed in this paper. We conducted the experiments on an Intel Xeon (R) 2.50GHz vCPU computer with 8.00 GB memory. To verify the performance of this method, we conducted quantitative and qualitative comparisons with the latest technologies of SnowflakeNet [30], SpareNet [31], PF-Net [11] and SCCR [32]. We evaluated our method on widely used public datasets PCN [33] and Stanford. The PCN and Stanford datasets are public point-cloud datasets that have greatly contributed to the research of 3D vision. In addition, we experimentally analyzed the point-cloud data that we had collected to fully prove the effectiveness of our method. Similar to [34], we used GPSNR and NSHD as quantitative evaluation indicators of the experiment. GPSNR [35] measures the error between Point Clouds A and B on the basis of the optimized peak signal-to-noise ratio (PSNR). The higher the GPSNR is, the smaller the difference between A and B. Normalized symmetric Hausdorff distance (NSHD) [36] is a normalized metric based on the unilateral Hausdorff distance. The lower the NHSD is, the smaller the difference between A and B.

### 3.1. Qualitative Analysis

Figure 5 shows the subjective repair effect of our and other comparison methods on the PCN dataset. We selected four types of objects in the PCN dataset: airplane, table, chair, and car. As can be seen from the results, our method could accurately repair the missing parts in the point cloud without producing redundant 3D points in nonmissing regions.

For example, in the table category, our method could focus on repairing the missing areas in the point cloud, keeping nonhole areas unchanged, and the distribution of newly filled points was more uniform than that of other methods. On the other hand, SpareNet could not focus on the missing area and only generated a complete point cloud on the basis of the trained parameter model. Therefore, redundant 3D points were generated in the nonpore area, blurring the original shape of the point cloud. The point cloud completed by the SnowflakeNet method had new missing areas on the chair point cloud because it had reconstructed the entire point-cloud model and generated new 3D points in the nonhole area. New hole areas appear when new 3D points are not evenly distributed. Our method predicts the surface shape of the missing area according to the distribution information of the points in the hole neighborhood in order to ensure that the distribution of 3D points in the nonhole area remains unchanged. PF-Net is a rough-to-fine process. To improve the resolution of the output results, the number of 3D points was increased. However, this is different from the ground truth. The SCCR method failed to completely repair the large holes on the table and the surface of the airplane because when the point distribution is too scattered, and the hole area is large, the SCCR method only generates new filling points on the basis of a small number of hole boundary points. In addition, our repair method achieved a better visual effect for the car model than that of related work.

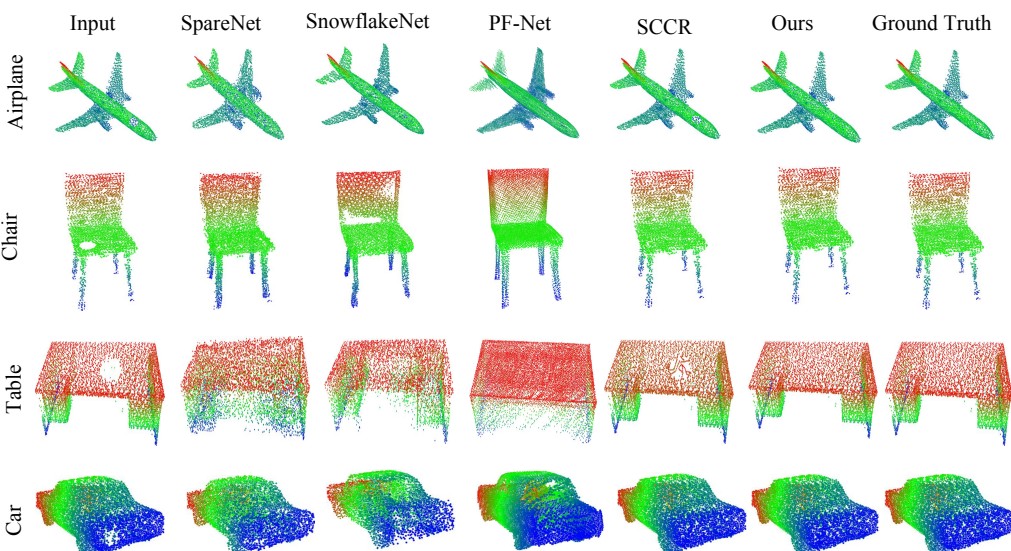

**Figure 5.** Subjective repair effect on PCN dataset.

Figure 6 shows the visualization effect on the point cloud collected by Stanford and our method. We selected monsters, monkeys, and a sphere as experimental point clouds, and none of these three categories of point clouds was used as training data for SpareNet, PF-Net, and SnowflakeNet. The monster point-cloud model was from the Stanford dataset, and we collected the monkey [37] and sphere point-cloud models. The experimental results show that SpareNet, PF-Net, and SnowflakeNet repaired the results, severely distorting the original shape of the point cloud and causing the repaired point cloud to be far from the true shape. SpareNet, PF-Net, and SnowflakeNet produced bad results because they were constrained by the training data and could not complete the untrained point cloud. However, training a large amount of data consumes huge amounts of time, so their methods cannot be widely used. Compared with the SpareNet, PF-Net, and SnowflakeNet methods, our method only focused on the missing areas and was not limited by the object category. At the same time, our method could effectively repair various point clouds regardless of target category. This shows that our method has wider applicability. On the pipe surface model, due to the large hole area and the scattered point distribution, the SCCR method could not completely repair the hole. Compared with the SCCR method, our method could

not only completely repair the holes in the point cloud, but also ensure that the repaired surface was consistent with the surrounding neighborhood.

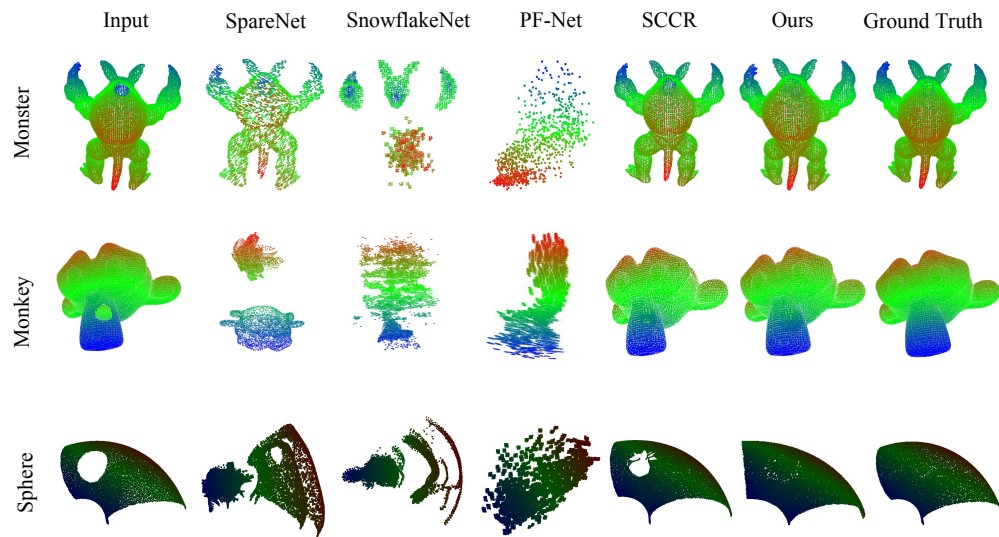

**Figure 6.** Effect of visualization on point cloud collected by Stanford [38] and our method.

Figure 7 shows the subjective repair effect of our method for multiple holes in a single point cloud. The rabbit point cloud is from the Stanford dataset, and there were many holes at the bottom. To prevent the influence of adjacent holes on the repair results, our method does not repair multiple holes at the same time, but repairs them one by one. The repaired surface was assumed to be a real surface that provided geometric information for repairing adjacent holes. Experimental results show that our method could repair multiple holes in a single point cloud and achieved satisfactory results.

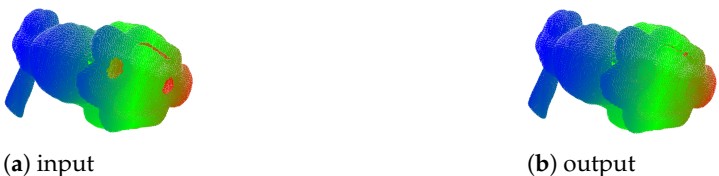

(**a**) input                    (**b**) output

**Figure 7.** Repair effect of multiple holes.

Figure 8 shows the repair effect of our method on irregular holes. Because the holes in this point cloud were naturally formed, there was no ground truth cloud with which to compare. The visualization results of the experiment show that our method could completely repair the point cloud and achieve ideal repair results.

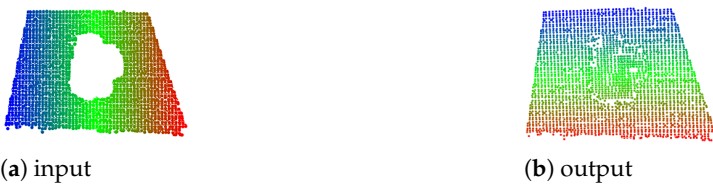

(**a**) input                    (**b**) output

**Figure 8.** Irregular-hole repair effect.

*3.2. Quantitative Analysis*

It is very important to objectively evaluate the geometric differences of point clouds. Similar to [5], we used GPSNR and NSHD as quantitative evaluation indicators of the experiment. Tables 1 and 2 show the quantitative comparison results between our method and related work. Experimental data show that our method achieved the best performance

in all categories. Specifically, Table 1 shows the quantitative comparison values of the normalized symmetric Hausdorff distance (NHSD), which is a normalized metric based on the unilateral Hausdorff distance. The lower the NHSD is, the smaller the difference between A and B. The experimental data show that our method achieved the lowest NHSD on each point-cloud model. On average, our method was 23.67 lower than the second method, with a relative decrease of 59.2%. SpareNet, PF-Net, and SnowflakeNet changed the 3D position in the source point cloud and generated redundant filling points in the nonhole area, so there was a large difference in the NHSD value between them and the ground truth. On the monkey, monster, and pipe surface models, the output of the SpareNet, PF-Net, and SnowflakeNet methods severely distorted the geometric shape of the point cloud (see Figure 6), so the values of NHSD of the SpareNet and SnowflakeNet methods were larger on these three types of point clouds. These results show that our method could repair point clouds with high stability and quality.

**Table 1.** Quantitative comparison of NHSD ($\times 10^{-2}$).

| Model | SpareNet [31] | SnowflakeNet [30] | PF-Net [11] | SCCR [32] | Ours |
|---|---|---|---|---|---|
| Airplane | 3.28 | 2.52 | 2.79 | 1.85 | **0.23** |
| Chair | 4.06 | 3.38 | 4.10 | 1.58 | **1.04** |
| Table | 18.56 | 13.39 | 19.31 | 4.25 | **0.51** |
| Car | 6.33 | 10.30 | 12.39 | 2.96 | **1.13** |
| Monster | 811.98 | 5567.67 | 6092.97 | 240.32 | **107.44** |
| Monkey | 290.11 | 529.26 | 603.14 | 2.77 | **1.21** |
| Sphere | 148.25 | 182.33 | 223.45 | 26.01 | **2.47** |
| Mean | 183.22 | 901.26 | 994.02 | 39.96 | **16.29** |

Table 2 shows the GPSNR performance comparison values of our and other methods. GPSNR is based on the optimized peak signal-to-noise ratio (PSNR) to measure the error between Point Clouds A and B. The higher the GPSNR is, the smaller the difference between A and B. The experimental data show that our method achieved the highest GPSNR on each point-cloud model with an average of 48.47 DB, which was 44.9% higher than that of a suboptimal method. The experimental results show that the point cloud repaired by our method had high fidelity.

**Table 2.** Quantitative comparison of GPSNR (DB).

| Model | SpareNet [31] | SnowflakeNet [30] | PF-Net [11] | SCCR [32] | Ours |
|---|---|---|---|---|---|
| Airplane | 18.54 | 19.31 | 18.01 | 40.38 | **62.41** |
| Chair | 15.15 | 14.48 | 15.11 | 32.05 | **37.71** |
| Table | 19.22 | 19.91 | 13.38 | 41.52 | **58.95** |
| Car | 15.48 | 14.53 | 13.51 | 40.20 | **44.23** |
| Monster | 4.81 | 5.01 | 2.33 | 32.41 | **51.72** |
| Monkey | $-2.52$ | 3.01 | $-3.74$ | 37.36 | **50.83** |
| Sphere | $-7.17$ | $-9.31$ | $-8.63$ | 10.17 | **33.45** |
| Mean | 11.78 | 9.56 | 7.14 | 33.44 | **48.47** |

## 4. Conclusions

This paper proposed a point-cloud repair method via a convex set for an incomplete point cloud. First, we constructed a cube unit $S$ centered on several adjacent hole boundary points and discretized $S$ into $K(K = 9)$ subunits. Second, pointless subunits that are roughly screened by the difference of normal vectors are called unstable unit sets. Then, the objective function was calculated from the joint distribution function of the attribute information of the unstable unit set and the state distribution function of MRF, and the MAP was solved by using variational inference to determine subunit $s^*$ that needed to be filled. Lastly, $s^*$ and central unit $s_1$ formed a specific subunit $s$. Combined with the weakened convex set, new 3D points were added to the hole area, so that subunit $s$ became a convex

set. Our method could accurately infer the distribution of new filling points to ensure that the repaired shape was as consistent with the real surface shape as possible. The proposed method was not limited by the category of point clouds and could repair various categories of point clouds while maintaining the position of points in non-hole areas unchanged. The experimental results show that our method is superior to the compared methods in terms of GPSNR and NHSD. However, the method in this paper was limited by the size of unit $S$. Large or small units $S$ reduce the final repair effect. Therefore, we will consider using more prior information to repair incomplete point clouds in the future.

**Author Contributions:** Conceptualization, T.D.; methodology, T.D. and Y.Z.; software, Y.Z.; formal analysis, Y.Z.; data curation, Y.B.; writing—original draft preparation, Y.Z.; writing—review and editing, M.L.; visualization, Y.B. and M.L.; funding acquisition, T.D. All authors have read and agreed to the published version of the manuscript.

**Funding:** This research was partially supported by the Shanghai Alliance Program (LM2019043).

**Institutional Review Board Statement:** Not applicable.

**Informed Consent Statement:** Not applicable.

**Data Availability Statement:** Not applicable.

**Conflicts of Interest:** The authors declare no conflict of interest.

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
