# Peer review of "Point Cloud Repair Method via Convex Set Theory"

_applsci, doi:10.3390/app13031830_

Round 1

Reviewer 1 Report

In this paper, the authors propose a Point Cloud Repair Method via Convex Set Theory. I have the following confusion in this paper, please explain and revise.

1. In the section of qualitative analysis, the authors' comparison between the details of the car shown by the SCCR method and the Ours method in Figure 4 is not obvious. The superiority of the proposed method cannot be shown.

2. The authors compare too few comparison methods. It is suggested to add the latest comparison methods to verify the performance of the proposed method in the experimental section.

3. The authors do not give the source of the dataset in Figure 5, and suggest introducing relevant references.

4. In the authors' quantitative comparison, it is suggested to show the best results in red and bold.

5. The authors should further refine the content of the conclusions.

6. It is suggested that the authors further refine and highlight the contributions made to the method of this paper.

7. The authors should carefully revise the grammatical issues of the article to make the article clearer for the readers.

Author Response

Reviewer #1:

  1. In the section of qualitative analysis, the authors' comparison between the details of the car shown by the SCCR method and the Ours method in Figure 4 is not obvious. The superiority of the proposed method cannot be shown.

Response 1: Sorry for the confusion. SCCR is an excellent point cloud repair method. The distribution of the car point cloud is scattered, and the geometric shape of the missing area is similar to that of the plane. Subjectively, SCCR can also show a good repair effect by adding enough 3D points in the hole area.

     However, the quantitative analysis in Table 1 and Table 2 shows that our method is superior to the SCCR method in the car model. The NHSD of SCCR is 2.96, more than twice that of our method (The lower the NHSD, the better the effect). Similarly, the GPSNR of our method is 44.23DB, which is 10% higher than the SSCR method (The higher the GPSNR, the better the effect).

  1. The authors compare too few comparison methods. It is suggested to add the latest comparison methods to verify the performance of the proposed method in the experimental section.

Response 2: We appreciate your contributional suggestions. We have added the latest comparison method to verify the performance of the proposed method in the experimental section (Section 3). We also added the state-of-the-art method PF-Net (CVPR, 2020) for comparison. PF-Net is a point cloud completion method based on deep learning, which can produce excellent repair results. In the revised version, we conduct detailed quantitative and qualitative analysis with four advanced point cloud repair methods.

Please refer to the revised Section 3 for details. 

  1. The authors do not give the source of the dataset in Figure 5, and suggest introducing relevant references.

Response 3: We also briefly introduced the datasets we used in the revised manuscript and added reference information. Please refer to the first paragraph of Section 3 of the revised manuscript. Thanks.

  1. In the authors' quantitative comparison, it is suggested to show the best results in red and bold.

Response 4: According to your suggestion, we show the best results in bold.

Please refer to the revised Table 1 and Table 2 for details.

  1. The authors should further refine the content of the conclusions.

Response 5: Thanks for the suggestion. We further refined the conclusions section and elaborated on the advantages, limitations, and future improvements of the proposed method.

Please refer to the revised Section 4 for details. 

  1. It is suggested that the authors further refine and highlight the contributions made to the method of this paper.

Response 6:  We first propose a point cloud repair method via convex set.  Considering the advantages of the convex set in dealing with uncertainty problems, we transform the point cloud repair problem into the construction of convex sets. We divide the hole edge into multiple subunits and introduce MRF to predict the special subunit that needs filling points. Then the special subunit is constructed into convex sets to gradually repair the entire hole area.

Our method is not limited by point cloud categories and has wide applicability. In addition, our method can accurately restore the original geometric features of the missing region and keep the non-hole region unchanged.

Please refer to the revised Section 1 for details. 

  1. The authors should carefully revise the grammatical issues of the article to make the article clearer for the readers.

Response 7: We apologize for the poor presentation.  We have tried to improve both language and readability. We hope the manuscript is more readable now.

Reviewer 2 Report

In this manuscript, the authors have considered repairing single holes from point cloud using MRF and MAP based method that they have been developed. Manuscript is well-written, with adequate number of relevant references included. Authors have clearly described the method they use  and comparison with other well-known methods have been presented adequately. Numerical results seems to be reasonable. There is missing indention in Table 1, which should be fixed. Also reference to the SpareNet is broken / missing. However, these are minor is issues, and I would recommend to publish this manuscript in the journal of Applied Sciences. 

Author Response

Reviewer #2

  1. There is missing indention in Table 1, which should be fixed. Also reference to the SpareNet is broken / missing.

Response 1:

Thanks for the suggestion. According to your request, we have corrected Table 1 and added the reference to SpareNet.

 Thank you very much for your approval of our article.

Reviewer 3 Report

In the study, a method for point cloud repair is proposed. The mathematical background of the method is well presented. However, the study needs major revision. Here are my suggestions:

1) The literature study should be expanded. Novelty should be revealed.

2) Combine Section 2 and Section 3. It will be more understandable if it has a flow under the heading Methods.

3) Are 3 adjacent points enough? Could you explain this a little more? How many adjacent points are required for an accurate assessment?

4) The expressions described in Section 3.2.2 are similar to the Superpoint Graph method. Was there such an inspiration when developing your method? If so, please indicate that as well.

5) The Methods section is too long. It would be appropriate to explain it more concisely. Especially Section 3.2.2 is too long. Please shorten these sections.

6) Brief information about the data sets used should be given.

7) There is an error in the reference of SpareNet in Table 1 and Table 2.

8) The discussion of your results is not sufficiently done. What makes your method superior to other methods? For example, why SnowflakeNet could not close the gap of the chair, but your method could? Please expand discussions like this. In this way, only the results are written. One of the most important parts of a scientific article is the discussion section.

Author Response

Reviewer #:

  1. The literature study should be expanded. Novelty should be revealed.

Response 1: 

    Thank you for your comment. We have expanded the literature study and revealed the novelty of the article in the revised version.

Please see the revised manuscript in Section 1.

  1. Combine Section 2 and Section 3. It will be more understandable if it has a flow under the heading Methods.

Response 2: 

To make logic clearer, we have combined Section 2 and Section3. The merged Section is named "Methods"(Section 2).

According to your suggestion, we added the flow chart of the proposed algorithm to the "Methods" for the readers to understand.

Please see the revised manuscript in Section 2; the flow chart is shown in Figure 2.

  1. Are 3 adjacent points enough? Could you explain this a little more? How many adjacent points are required for an accurate assessment?

Response 3: 

Sorry for the confusion. We have revised it in the manuscript. To improve the rigor of the article, we changed "3 adjacent points" to " adjacent points".

The size of  is positively correlated with the density of adjacent points, and the greater the density, the smaller the distance between adjacent boundary points. Therefore, more boundary points are needed to construct an appropriate size of regional space. When the density is smaller, the distance between adjacent boundary points is larger, so fewer boundary points are required.

In our article, to avoid too small regional space, we choose .

  1. The expressions described in Section 3.2.2 are similar to the Superpoint Graph method. Was there such an inspiration when developing your method? If so, please indicate that as well.

Response 4: Section 3.2.2 has now been adjusted to Subsection 2.3.2. The expression described in this Subsection is Markov Random Field (MRF) from the undirected graph model, which is a random process. The core idea of this model is that the value of each node is only related to the surrounding nodes.

Although Superpoint can also make the local domain features of point clouds consistent, our method considers the Markov property and is not inspired by Superpoint Graph. We can learn the idea of Superpoint Graph, which will help our work in the future. We appreciate your contributional suggestions.

  1. The Methods section is too long. It would be appropriate to explain it more concisely. Especially Section 3.2.2 is too long. Please shorten these sections.

Response 5:  We have shortened the contents of relevant sections to improve the readability of the article. We have integrated some formulas in the revised manuscript and deleted or adjusted some statements. Please refer to the revised Section 2 for details. 

  1. Brief information about the data sets used should be given.

Response 6: PCN and Stanford data sets are public point cloud data sets. They have made significant contributions to the research of 3D vision. Scholars who study point cloud segmentation and point cloud repair often verify the effectiveness of their methods on PCN data sets.

We also briefly introduced the datasets we used in the revised manuscript and added reference information. Please refer to the first paragraph of Section 3 of the revised manuscript.

  1. There is an error in the reference of SpareNet in Table 1 and Table 2.

Response 7:

Thanks for the suggestion. We have corrected the errors in Table 1 and Table 2.

  Please refer to Table 1 and Table 2 for revision details. We are sorry for the errors caused by our negligence.

  1. The discussion of your results is not sufficiently done. What makes your method superior to other methods? For example, why SnowflakeNet could not close the gap of the chair, but your method could? Please expand discussions like this. In this way, only the results are written. One of the most important parts of a scientific article is the discussion section.

Response 8:

   Thanks. According to your suggestion, we supplemented the discussion. We analyzed and explained the reasons for the lack of relevant methods, and compared them with our methods to conclude.

For example, because the SnowflakeNet will reconstruct the entire point cloud model and generate new 3D points in the non-hole area, SnowflakeNet cannot close the gap of the chair. New hole areas are displayed when new 3D points are not evenly distributed. Our method predicts the surface shape of the missing area according to the distribution information of the points in the hole neighborhood, to ensure that the distribution of 3D points in the non-hole area remains unchanged.

Please refer to the revised Section 3 for details. 

Reviewer 4 Report

Dear authors! Your article was very interesting to me. It was a pleasure to read and meet her. You have developed a wonderful method and in my opinion applicable in practice. I wish you health and fruitful work in the future!

Author Response

Thank you very much for your approval of our article.

Round 2

Reviewer 3 Report

The authors made the suggested revisions. The work appears ready for publication as such.